# Design, Analysis, and Experimental Evaluation of a New Secure Rejoin Mechanism for LoRaWAN Using Elliptic-Curve Cryptography

Stefano Milani [†] and Ioannis Chatzigiannakis *,[†]

Department of Computer, Control, and Management Engineering "Antonio Ruberti", Sapienza University of Rome, 00185 Rome, Italy; milani.1707181@studenti.uniroma1.it
* Correspondence: ichatz@diag.uniroma1.it
† These authors contributed equally to this work.

**Abstract:** LoRaWAN (Long Range Wide Area Network) is a Low-Power Wide Area Networks (LPWAN) technology with very rapid uptake during the previous years, developed by the LoRa (Long Range) Alliance as an open standard operating over the unlicensed band. Current LoRaWAN architecture foresees specific techniques for bootstrapping end-to-end encryption during network initialization. In particular, this work focuses on the *Over-The-Air Activation* (OTAA) method, which uses two keys (Network key (`NwkKey`) and Application key (`AppKey`)) that are hard-coded into the device and do not change throughout the entire lifetime of the deployment. The inability to refresh these two keys is as a weak point in terms of the overall security of the network especially when considering deployments that are expected to operate for at least 10–15 years. In this paper, the security issues of OTAA are presented in detail highlighting the vulnerabilities against the specific type of attacks. A new scheme for network activation is proposed that builds upon the current LoRaWAN architecture in a way that maintains backwards compatibility while resolving certain vulnerabilities. Under the new mechanism, the devices periodically negotiate new keys securely based on elliptic-curve cryptography. The security properties of the proposed mechanism are analyzed against a specific type of attacks. The analysis indicates that the new secure rejoin mechanism guarantees (i) computational key secrecy, (ii) decisional key secrecy, and (iii) key independence, forward and backward, for both root keys thus properly addressing the considered security vulnerabilities of LoRaWAN. Moreover, the method is implemented in software using the RIOT-OS, a hardware-independent operating system that supports many different architectures for 8 bit, 16 bit, 32 bit and 64 bit processors. The resulting software is evaluated on the FIT IoT-Lab real-world experimentation facility under a diverse set of ARM Cortex-M* devices targeting a broad range of IoT applications, ranging from advanced wearable devices to interactive entertainment devices, home automation and industrial cyber-physical systems. The experiments indicate that the overall overhead incurred in terms of energy and time by the proposed rejoin mechanism is acceptable given the low frequency of execution and the improvements to the overall security of the LoRaWAN1.1 OTAA method.

**Keywords:** network activation; key agreement; low-power long range networks; internet of things; network security; protocol design; performance evaluation; RIOT OS; real-world experimentation



## 1. Introduction

The need to deliver scalable and long-term networks of devices that integrate sensing, computation and wireless communication in small, low-power devices that can be seamlessly embedded in complex physical indoor and outdoor environments has given rise to sub-GHz wireless communication [1,2]. The approach of trading-off data transmission rate while keeping power consumption at low levels has created the so-called Low-Power Wide Area Networks (LPWANs) [3]. In contrast to more high-frequency communication,

low-frequency signals are not as attenuated by thick walls or multipath propagation as high-frequency signals contributing in this way to robustness and reliability of the signal [1]. Overall, LPWANs are considered promising candidates to deliver urban-scale coverage in the context of smart city services [4–6] or deployments over rural environments, since they allow *high energy autonomy* of the connected devices, *low device and deployment costs*, *high coverage capabilities* and support *large number of devices* [7,8].

LPWAN technologies allow embedded devices to wireless communicate directly to gateways (also called a *collector* or *concentrator*) over distances in the range of several kilometres. Although this single-hop connectivity simplifies communication, it exposes message exchanges to malicious nodes that could be potentially located at long distances. Recent studies concentrate on the security vulnerabilities in LPWANs [9,10] providing alternative solutions for the used cryptographic primitives [11,12], e.g., the authors in [13] focus on application server vulnerabilities and in [14] it is introduced an alternative key management scheme. An experimental evaluation of the performance of Long-Range Low-Power Wireless Communication is carried out in [15] that highlights the limitations of the new technology. In [16] a hierarchical structure is proposed that combines LPWANs with short-range IEEE 802.15.4 networks to reinforce network security [16]. The possibility of introducing a Median Server to reinforce the trustworthiness of the network is proposed in [17].

This paper looks into LoRaWAN (Long Range Wide Area Network), an LPWAN technology developed by the LoRa (Long Range) Alliance as an open standard [18] that can operate over private and/or public infrastructures spreading all over the world [19]. In particular, the focus is on the way nodes join the network under the *Over-The-Air Activation* (OTAA) method of LoRaWAN, and how the key agreement protocol between the end-devices and the Network/Join server can be improved in terms of security. Under the latest LoRaWAN specifications v1.1 (LoRaWAN1.1), OTAA uses two root keys, the Network key (`NwkKey`) and the Application key (`AppKey`), to compute the session keys. While OTAA allows refreshing the session keys, it is not possible to refresh the aforementioned root keys. These two keys are hard-coded into the device firmware and remain the same for the entire duration of the network deployment. This is a vulnerability of OTAA taking into account that a device may be expected to be operational for at least ten years. The paper presents a series of security issues of OTAA and exposes specific vulnerabilities connected to passive Man In the Middle and Replay attacks.

The solution proposed in this paper builds on top of the existing version of OTAA without modifying the architecture of the LoRaWAN1.1 specification thus ensuring backwards compatibility. In particular, the proposed solution uses the message structure defined in LoRaWAN1.1 specification thus devices already deployed will continue to operate without any disruption. It is, therefore, possible to deploy devices with a new version of OTAA that implements the mechanism presented here without disrupting the operation of devices deployed with the standard version of OTAA. The new mechanism introduces a new type of *Rejoin-Request* and *Join-Accept* message that allow refreshing the root keys using *Elliptic-Curve Cryptography* (ECC) in a way that guarantees *computational key secrecy*, *key secrecy*, *key independence* and *key freshness* [20–22]. Elliptic-Curve Cryptography was first in 1986 [23] and since then it has widespread exposure and acceptance in a wide variety of applications [24]. Recently, ECC-based security solutions have been introduced also in the case of low end IoT and embedded devices with constrained execution environments [25]. Remark that the solution proposed here is using an asymmetric encryption scheme based on ECC where a pair of public key and a private key are used to encrypt and decrypt messages when communicating. On the other hand, the current version of the LoRaWAN1.1 specification is using a symmetric encryption schema based on AES where only one key is used to encrypt and decrypt information.

The proposed mechanism is implemented using the *RIOT Operating System* [26], a multi-purpose operating system for IoT deployments, based on well-established crypto-libraries for elliptic-curve cyrptography [27]. The resulting firmware was evaluated using

the real-world experimentation facility of FIT IoT-LAB testbed [28]. The experimental evaluation was based on a heterogeneous set of microcontrollers (MCU) based on the ARM Cortex-M architecture group, ranging from low power M0/M0+ version up to M4/M4F more powerful versions. The Arm Cortex-M is group of a 32-bit RISC ARM processor cores optimized for low-cost and energy-efficient integrated circuits. These cores have been embedded in tens of billions of consumer devices targeting a broad range of IoT applications, ranging from ultra-low power wearable devices to interactive entertainment devices, home automation and industrial cyber-physical systems. The experimental evaluation indicates that the time and energy efficiency of the proposed mechanism and the overall overhead incurred is acceptable given the low frequency of execution and the improvements to the overall security of the LoRaWAN1.1 OTAA method.

The rest of the paper is organized as follows. In Section 2 the previous and related work is presented. A detailed presentation of the Over-The-Air Activation (OTAA) under the latest LoRaWAN specifications v1.1 (LoRaWAN1.1) is presented in Section 3 while in Section 4 the weak points and possible vulnerabilities of the activation method, referring to different attack models, are discussed. The new Secure Rejoin Mechanism is presented in Section 5. The analysis of the security properties of the new mechanism is discussed in Section 6 and in Section 7 the performance of the method is evaluated in terms of time and energy efficiency by carrying out a series of real-world experiments. Finally, in Section 8 the conclusions and future work direction are presented.

## 2. Previous and Related Work

The security of LPWANs is an open issue, and several works analyze possible vulnerabilities. An important aspect of security is the heterogeneity of the end-devices that can be connected to LPWANs, magnifying security threads with respect that to the current internet and making it crucial to set high standards of security, privacy, and trust [29]. A detailed analysis of LoRaWAN security, ranging from the hazard of physical access to the end-device to the possibility to perform an ACK spoofing attack or an Application-Specific attack is presented in [30]. The analysis indicates how through eavesdropping one can compromise the encryption method and manage to decrypt part or even the entire cypher-text by employing frame counter resets when the session keys remain the same. The analysis underlines the importance of having an efficient and complete key refresh mechanism in every Key Agreement Protocol.

Regarding LoRaWAN1.1 specifically, some potential vulnerabilities are presented in [31]. They focus on the possibility of performing replay attacks relaying on jamming techniques, to lead to a Denial Of Service. They also point to a vulnerability to the non-secured beacons in the case of LoRaWAN class B devices to de-synchronize the receive windows and the possibility of network analysis. In [10] the combination of a jamming attack to a replay attack is proposed. Additionally, they focus on a replay attack in the case of a badly configured application server, i.e., when the frame counter is disabled. In [17] is proposed a new architecture for LoRaWAN networks, to reinforce security and to provide an end-to-end secure communication scheme. The authors point as a possible solution a Median Server, a new entity in the architecture that has the role of a registration authority for both end-devices and gateways. To fulfil this purpose a Central Authority is introduced to ensure that only authenticated devices interact with the system and that they connect only with authenticated gateways.

The method presented here does not modify the LoRaWAN1.1. specification but rather relies on *Elliptic-Curve Cryptography* (ECC) [32,33] to allow the end-device and the Join/Network server to agree on the new pair of root keys. ECC allows having the same level of security of RSA using smaller parameters [34] for example an elliptic curve over a 283-bit field gives the same level of security as a 3072-bit RSA modulus or Diffie-Hellman prime. As is shown in [35,36], ECC is a better alternative concerning RSA for resource-constrained devices used in IoT context, both in terms of computational power and energy

efficiency. In particular, in [35] it is shown how using ECC add a small run-time overhead that is worth the gain in terms of security.

In terms of designing key agreement protocols for resource-constrained devices, the authors in [37] perform an attack exploiting a vulnerability in the IEEE 802.15.4 ZigBee OTAA and demonstrate how to update the firmware of smart lights with a malicious code that spreads over other smart bulbs. This shows the importance of a secure activation mechanism, both in the network initialization phase and in the phase of updating the firmware or the keying material. A lightweight bootstrapping protocol for authentication and establishing credentials for 4G/5G and Narrow-Band IoT (NB-IoT) networks is presented in [38]. The proposal finds LO-CoAP-EAP as a feasible and efficient solution for NB-IoT and 5G networks. For wireless sensor networks, in [20] a Group Key Establishment protocol is proposed that does not require many-to-many messages. In [39] an agent-based key establishment protocol is proposed that does not rely on a global ordering of devices or the construction and maintenance of a distributed structure that reflects the topology of the network. These protocols are using the Elliptic-Curve Diffie-Hellman protocol to generate a shared secret between the devices. Elliptic Curve Cryptography is also used in [40] that propose two group key agreement protocols for enabling a secure multi-casting in Wireless Sensor Networks (WSNs). Both protocols permit using asymmetric encryption at a low cost in terms of energy consumption and computational power.

Regarding LoRaWAN OTAA, in [41] the authors propose some alternatives to the actual key management in LoRaWAN. They explore the possibility of an approach based respectively on Internet Key Exchange version 2 (IKEv2), Datagram Transport Layer Security (DTLS) and Ephemeral Diffie-Hellman Over COSE (EDHOC), analysing the pros and cons of each approach, but they do not formalize an alternative key management scheme for LoRaWAN using one of the aforementioned protocols. In [42] it is proposed a root key distribution scheme for LoRaWAN OTAA. The authors make use of Rabbit, a high-efficient synchronous stream cypher, to generate a new pair of root key. After the generation of the new keys, the end device will trigger a new Join-Request with the Join Server. The authors in [43] propose a new key management schema for LoRaWAN based on *Hierarchical Deterministic Wallet* using the BIP32 algorithm [44].

In contrast to the aforementioned solutions, the method presented here allows refreshing the root keys without any modification of the actual architecture of LoRaWAN1.1. It provides an efficient way to generate a new pair of root keys by exchanging only two messages. The analysis performed indicates that certain vulnerabilities of LoRaWAN1.1 can be improved without incurring significant overhead in the overall performance of the network in terms of time and energy efficiency.

## 3. Over-The-Air Device Activation in LoRaWAN1.1

LoRa is a physical communication layer, using the proprietary modulation technique LoRa Modulation, a derivative of the Chirp Spread Spectrum (CSS), which operates in the Sub-GHz bands and is developed and distributed by Semtech [45]. LoRaWAN allows end-nodes to communicate independently and asynchronously, similarly to an ALOHA protocol with bit rates varying from 100 bps to around 5.5 kbps. A LoRa concentrator can receive data on the same channel from multiple end-devices at the same time if the bit rates are different.

In LoRaWAN the encryption of the payload is by default enabled in every transmission. The data frame of an end-node has a 32-bit identifier, a 7-bit network identifier and a 25-bit network address and the maximum payload is 250 bytes. Since end-devices are not assigned to a specific concentrator, the data frames do not include any concentrator identifier. In this way, anyone can receive the encrypted data packets. To prevent replaying packets, a frame counter is used both for upstream and downstream messages which will block transmission from being sent more than once.

Two different 128-bit AES keys are used for a two-step message chain for both upstream and downstream message exchanges. In the first step, the Application Session Key

(*AppSKey*) is used to encrypt the data frame between the end device and the application server. In the second step, a Network Session Key (*NwkSKey*) is used to verify the authenticity of the nodes. The data frame exchanged between the end-device and the Join/Network server is encrypted with the *NwkSKey*.

This work focuses on the *Over-The-Air Activation—OTAA* method that permits the end-device and the Join/Network Server to agree upon the session and the integrity keys using a handshake process [45]. OTAA makes use of some hard-coded keys that are used to determine the needed keys.

### 3.1. Prerequisites for Device Activation Using OTAA

The OTAA method assumes that each device is uniquely identified by a 64 bit network-wide identifier. The so-called `DevEUI` must be stored into the device before the network activation. It can be public, and it is a recommended practice to make it available on the device label.

Similarly, a cloud-based application that communicates with end-devices is also uniquely identified by a 64 bit network-wide identifier. The so-called `JoinEUI` must also be stored in the device before network activation.

In addition to the above identifiers that are encoded in the end-device and defined in the Join/Network Server, both parties agree on a common set of keys before the network initialization phase. These are the *Device Root Keys*, the `NwkKey` and the `AppKey`, two AES-128 root keys. These keys are the ones used to derive the session and integrity keys. They never change during the life-cycle of the end-device, so securing distribution, storage and usage of them, both in the end-device and in the Join/Network Server, is crucial for providing end-to-end network security and guaranteeing the confidentiality of data. Finally, based on the `NwkKey`, the end-device is using two additional keys:

`JSIntKey` used to compute the Message integrity code (MIC) for the Rejoin-Request message and the Join-Accept message as follows:

$$JSIntKey = aes128\_encrypt(\texttt{NwkKey}, 0x06|DevEUI|pad_{16})$$

`JSEncKey` used to encrypt the Join-Accept message triggered by a Rejoin-Request.

$$JSEncKey = aes/_e ncrypt(\texttt{NwkKey}, 0x05|DevEUI|pad_{16})$$

### 3.2. Initiating Device Activation Using the Join-Request Message

The device activation method is initiated with the end-device transmiting an up-stream *Join-Request message*. The message contains the `JoinEUI`, the `DevEUI`, and a `DevNonce` that is a counter, starting at 0, incremented at each *Join-Request message*. The counter is introduced as a countermeasure for the replay attack since it cannot be used twice for a given *JoinEUI*. The *Join-Request message is NOT encrypted*. The format of the *Join-Request message* is depicted in Table 1. The *MIC* for the message is computed as follows:

$$cmac = aes128\_cmac(\texttt{NwkKey}, MHDR|JoinEUI|DevEUI|DevNonce)$$

$$MIC = cmac[0...3]$$

**Table 1.** Join-Request packet format.

| JoinEUI | DevEUI | DevNonce |
|---------|--------|----------|
| 8 Bytes | 8 Bytes | 2 Bytes |

### 3.3. Device Activation Using the Join-Accept Message

The Join/Network server upon accepting the *Join-Request message*, verifies the identifiers and decides if the end-device should be accepted. If the end-device is accepted,

the Join/Network server computes the `DevAddr`, a 32-bit identifier for the end-device within the LoRaWAN network. The Join/Network server sends a down-stream *Join-Accept message* to the device contains the `DevAddr`, a network identifier `Home_NetID` and the operational parameters of the network (`DLSettings`, `RxDelay`, `CFList`). The message also includes the `JoinNonce`, a device-specific counter that never repeats itself, used by the end-device to derive the session and integrity keys. It is incremented at each *Join-Request message*. The format of the *Join-Accept message* is depicted in Table 2.

**Table 2.** Join-Accept packet format.

| JoinNonce | Home_NetID | DevAddr | DLSettings | RxDelay | CFList |
|:---:|:---:|:---:|:---:|:---:|:---:|
| 3 Bytes | 3 Bytes | 4 Bytes | 1 Byte | 1 Byte | 16 Bytes (optional) |

The end-device after receiving the down-stream *Join-Accept message* it computes the following session keys (for a graphical representation see Figure 1):

Forwarding Network Session Integrity Key—`KNwkSIntKey`—used by the device to calculate the *MIC* for all up-link messages.

$$FNwkSIntKey = aes128\_encrypt(\texttt{NwkKey}, 0x01|JoinNonce|JoinEUI|DevNonce|pad_{16})$$

Serving Network Session Integrity Key—`SNwkSIntKey`—used by the end-device to check the *MIC* of all down-link messages.

$$SNwkSIntKey = aes128\_encrypt(\texttt{NwkKey}, 0x03|JoinNonce|JoinEUI|DevNonce|pad_{16})$$

Network Session Encryption Key—`NwkSEncKey`—used to encrypt and decrypt both up-link and down-link MAC commands transmitted as payload on *port* 0 or in the *FOpt field*.

$$NwkSEncKey = aes128\_encrypt(\texttt{NwkKey}, 0x04|JoinNonce|JoinEUI|DevNonce|pad_{16})$$

Application Session Key—`AppSKey`—used by both the device and the Application Server to encrypt and decrypt the payload field of application-specific messages.

$$AppSKey = aes128_encrypt(\texttt{AppKey}, 0x02|JoinNonce|JoinEUI|DevNonce|pad_{16})$$

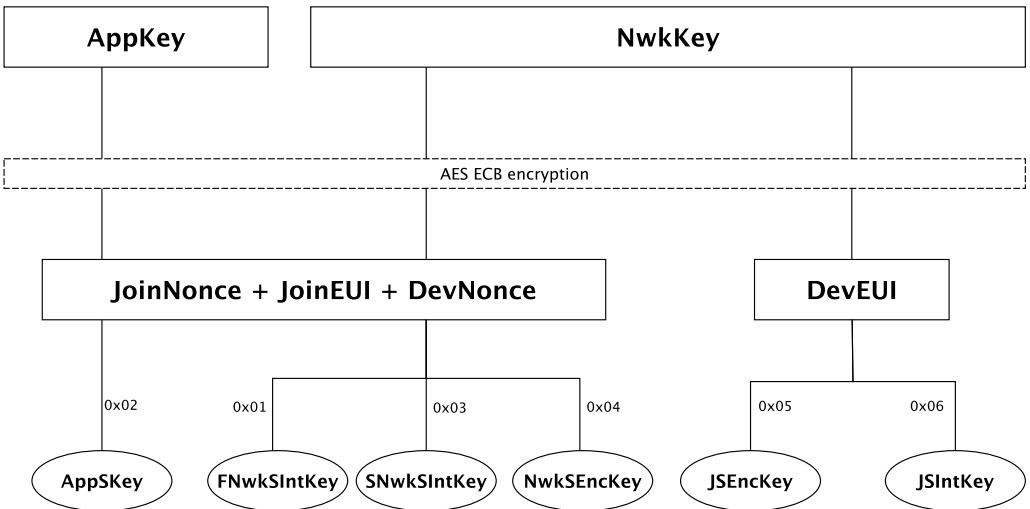

**Figure 1.** LoRaWAN 1.1 Key Derivation scheme [45].

*3.4. Refress Session Keys Using the Rejoin-Request Message*

Throughout the lifetime of an end-device, the session keys computed during the device activation can be periodically refreshed. In such a case, the end-device sends an upstream *Rejoin-Request message* to the Join/Network server. The *Rejoin-Request message is NOT encrypted*. Three types of *Rejoin-Request* message exist:

Rejoin-Request type 0—used to reset a device context including all four session keys. It contains the `NetId`, the `DevEUI` and the `RJcount0` counter that is incremented at every transmission of *Rejoin-Request message of type 0*. If the *RJcount0* reaches $2^{16} = 1$ the device shall stop transmitting Rejoin-Request of the given types, and restart device activation method using a *Join-Request message*. The format of the *Rejoin-Request message of type 0* is depicted in Table 3.

Rejoin-Request type 1—is similar to to type 0 but transmitted on top of normal application traffic without disconnecting the end-device. It contains the `JoinEUI`, the `DevEUI` and the `RJcount1` counter that is incremented at every transmission of a *Rejoin-Request message of type 1*. The `RJcount1` counter shall never warp around, due to the lifecycle of the device for a given *JoinEUI* value. The format of the *Rejoin-Request message of type 1* is depicted in Table 4.

Rejoin-Request type 2—is to update the 32 bit identifier of the end-device within the LoRaWAN network `devAddr` while keeping the same radio parameters. The message format is the same with the *Rejoin-Request message of type 0* apart from the `RJcount0` that is replaced by `RJcount2` (see Table 3). The `RJcount2` is incremented at every *Rejoin-Request message of type 2* and when it reaches $2^{16} = 1$ the device shall restart device activation method using a *Join-Request message*.

**Table 3.** Rejoin-Request message of type 0 or type 2.

| Rejoin Type = 0 or 2 | NetID | DevEUI | RJcount0 |
|---|---|---|---|
| 1 Byte | 3 Bytes | 8 Bytes | 2 Byte |

**Table 4.** Rejoin-Request type 1 message.

| Rejoin Type = 1 | JoinEUI | DevEUI | RJcount1 |
|---|---|---|---|
| 1 Byte | 8 Bytes | 8 Bytes | 2 Byte |

For all the three types of the *Rejoin-Request message* the Join/Network Server responds by transmitting a down-link *Join-Accept message* to modify the device's network identity. The *RJcount0* or the *RJcount1* replaces the *DevNonce* in the key derivation.

## 4. Security Issues & Vulnerabilities of LoRaWAN1.1 Over-The-Air-Activation

In LoRaWAN1.1, the keys used for securing communication are stored by the end-devices, the Network Server and the Application Server. If one of the Network/Application servers is compromised the attacker could view and modify all the communication with the corresponding end-devices. This paper looks into the Over-The-Air activation of end devices, hence the proposed mechanism does not look into how the keys are stored within the Network/Application Servers. Moreover, no mechanism is proposed to reinforce the overall security of these central elements of the architecture. Instead, the proposed method looks into the way the end-device and the Join/Application server can generate a new pair of keys in a secure way that overcomes certain types of attacks while maintaining specific properties.

When examining the possible vulnerabilities of LoRaWAN1.1 and in particular the Over-The-Air-Activation method one needs to take into consideration the goals of the attacker [9]. Since this paper looks into the key agreement process between the end-device

and the Join/Application Server, it is assumed that the goal of a potential attacker is to collect any information that will be useful when trying to guess the session and integrity keys used to encrypt the communication between the end-devices and the application server. An attacker that succeeds in guessing the session keys can decrypt the messages, impersonate the end device and in general carry out various active or passive attacks for the particular period where the device remains active. An attacker that succeeds in guessing the root keys can decrypt all the messages and in general impersonate the end-device until the end of its lifetime.

Since message authentication code is produced using the `NwkSkey` which is confirmed by the Network server, the Network server needs to be a trusted element to perform this task properly. However, both the Network Server and the intermediate concentrator (or an attacker on the intermediate network) are in a position to modify the encrypted payload without the Application server being able to notice the change. An adversary possessing the session key can generate a LoRaWAN message that will pass the signature checking procedure at the network server.

Another consideration related to certain application scenaria is connected to the fact that end-device may be placed in unprotected locations and are expected to operate for a very long time. In such deployments, it may be impractical and/or costly to increase the physical security level of the end devices for the complete duration of operation of the network. It is, therefore, possible that an adversary may manage to extract or acquire access to a small number of end devices to retrieve the secret keys. Although this paper does not look into how to reinforce the physical security of the device, it is, however, important to guarantee that the theft of keys from one end-device does not compromise other end-devices in the network.

The encryption mechanism used in LoRaWAN1.1 relies on AES operating in counter mode (CTR). In this mode of operation, end-devices generate ciphertexts which are the output of the XOR procedure on the string that contains a counter, the `AppSkey` and the plaintext (for a more detailed description of AES encryption in counter mode see [46]). This mode of operation is selected to reduce energy consumption and minimize the overall message encryption delays. As a result, encryptions are vulnerable to chosen ciphertext attack since if an attacker changes the payload data she can figure out which bit position in the encrypted payload corresponds to the same bit position in the plaintext (see [47]). On top of this, since authentication control is handled by the network server, the application server cannot verify the authenticity of the messages.

Assuming that an attacker manages to collect the root keys hard-coded into the end-device during fabrication, LoRaWAN1.1 does not provide a mechanism to change these keys during the entire life-cycle of the device. Therefore, the only way to refresh these keys is to manually change them, both in the Network/Application Server and into the end device. In such a case, the end-devices need to be reached physically, a process that may be expensive in application scenaria where end-devices are placed in remote or locations that are hard to reach.

A third element that needs to be examined regarding the effective security measures used in existing LoRaWAN deployments has to do with the protection against the tampering of the concentrators. In current deployments, it is fairly easy to take over some concentrators participating in the LPWAN and monitor all traffic passing through this point of the deployment. An attacker can record, replay and potentially manipulate the traffic passing through the concentrator. In the solution proposed in this paper, concentrators do not participate in the establishment of the security keys and the mechanisms introduced assume that only the end-device and the Join/Application servers can be considered trustworthy.

Looking into the Over-The-Air-Activation phase of the network, an attacker performing a simple passive Man In The Middle Attack that eavesdrops on messages exchanged by an end device can read any encrypted contents. Such an attacker can collect the following information that may be useful for an attacker to compromise the session and the integrity

keys are the `NwkKey`, `AppKey`, `JoinNonce`, `JoinEUI`, and the `DevNonce`. In particular, when an end-device transmit a Join-Request message since it is not encrypted the attacker collects the `JoinEUI` and the `DevNonce`. This information is always available to the attacker even in the case of a Rejoin-Request message since also this message is unencrypted. From a Rejoin-Request of type 0 or 2, an attacker can extract the `DevEUI`, and the `RJcount0`. From a Rejoin-Request message of type 2 instead, the attacker can collect the `JoinEUI` and the `RJcount1`. The `RJcount0` or the `RJcount1` take the place of the `DevNonce` during a rejoin request. So an attacker knows the `DevNonce` and the `JoinEUI` at each time of the life-cycle of the device. Remark also that the `JoinNonce` is a device-specific counter incremented at each Join-Request message, so it is easy for the attacker to compute the value of this information by counting the number of Join-Request the end-device send to the Network Server.

Another potential vulnerability of the Over-The-Air-Activation phase of the network is related to potential replay attack as pointed out in [31]. Using a selective RF jamming technique an attacker can potentially jam and capture the first Join-Request message transmitted by the end device. Given that the jamming attack will block the message from arriving at the Network server, no Join Accept message will be generated. After a fixed timeout period the device will be forced to re-transmit a second Join-Request message. Assuming that the attacker will succeed in jamming the second Join-Request message as well, it will now transmit the first jammed Join-Request message to the Network server. In such a scenario, the Network Server will respond to the first Join-Request with a Join Accept message. At this point and on, the Network Server, the Join Server and the end-device are de-synchronized.

## 5. A New Secure Rejoin Mechanism

In this section, the details of the proposed Secure Rejoin mechanism are presented along with technical details that support and motivate the choices made. Given the architecture of LoRaWAN1.1 and the fact that the protocol is already rolled out and used in operational environments, the proposed mechanisms are backwards compatible. This is achieved by maintaining the architectural elements of LoRaWAN and thus the solution proposed here is backwards compatible with the three types of Rejoin-Request described in Section 3.4. Moreover, the new mechanism follows the philosophy of LoRaWAN1.1 and requires the exchange of only two messages, a new type of *Rejoin-Request message* and a respective new type of *Join-Accept message*, to refresh all the keying material, including the root keys.

The proposed mechanism refreshes the LoRaWAN root keys using the *Elliptic-Curve Diffie-Hellman* protocol (ECDH) [48]. The *ECDH* is is similar to the classical Diffie-Hellman Key Exchange protocol but it uses *Elliptic-Curve Cryptography* (ECC) and specifically *ECC point Multiplication* instead of *modular exponentiation* [49].

The ECC curves are implemented over two number fields, the prime field, and the binary field. The *binary field curve* has worse performance and energy efficiency when executed on general-purpose processors [36], but outperforms the *prime field curve* if dedicated hardware is used. Since the end-devices are heterogeneous and in most cases are cheap boards with no particular hardware capabilities, it is recommended to use a prime field curve to implement the new type of Rejoin. In this way, good time and energy efficiency can for all devices. For a detailed evaluation of the performance of the proposed mechanism for a heterogeneous set of end-devices, see Section 7. In particular, the `secp256k1` and the `secp256r1` curves as the most suitable for low-power end-devices since smaller curves do not guarantee always better performances [36] while the key size of these two curves fit the purpose of the proposal.

The main difference between the two candidate curves is that the *secp258k1* curve is generated over a prime field associated with a *Koblitz curve*, on the other hand, the *secp256r1* curve is generated over random domain parameters. *Koblitz curves* are generally less secure but in the 256-bit curve, the impact is minimal [50]. The *secp256k1* curve is generally faster than the other curve if the implementation is optimized, in particular for the signature

generation and verification, and that is one of the motives the curve is used by Bitcoin [51]. On the other hand, the elapsed time difference for public key and secret generation in the two types of curves is very low [52]. Since the differences in terms of time and energy efficiency are negligible, the *secp256r1* curve is selected as it offers a higher level of security.

Based on the above choice of ECC curve, the *ECDH* protocol works as follows:

1. The end-device (*A*) and Join/Network server (*B*) have to agree on a common *Elliptic-Curve Group G* of order *n*, and on a *Primitive Element* $P \in G$ of order *n*. The *secp256r1* curve defines these parameters.
2. *A* selects an integer $a \in [2, \, n-1]$ and it computes $Q = [a]P$. Where *Q* is the public key of *A*, and *a* is the private key.
3. *B* selects an integer $b \in [2, \, n-1]$ and computes it $R = [b]P$. Where *R* is the public key of *B*, and *b* the private key.
4. The two parties exchange their public keys.
5. *A* computes the secret $S_A = [a]R = [a][b]P$
6. *B* computes the secret $S_B = [b]Q = [b][a]P$
7. The two parties have a common secret $S_A = [a]R = [a][b]P = [b]Q = S_B$

### 5.1. Initiate the Root Key Refresh Mechanism Using a Rejoin-Request Message of Type 3

The process starts with the end-device generating a new pair of public/private ECC keys using the *secpr256r1* curve. A new type of *Rejoin-Request message* is used that permits refreshing all the key materials at once. The *Rejoin-Request message of type 3* uses one byte for the `RejoinType`, which is set to 3; the `NetID`; the `DevEUI`; the `RJcount3`, which behaves in the same way of the `RJcount0` and `RJcount1`, and it will substitute the `DevNonce` for the subsequent session key generation; and the previously generated *compressed public key* of the device. Following the LoRaWAN1.1 OTAA specification, the *Rejoin-Request message is NOT encrypted*. The contents of the new packet type are depicted in Table 5. The `Message Integrity Code` of the Rejoin-Request message of type 3 is computed as follows:

$$cmac = aes128\_cmac(SNwkSIntKey,$$

$$MHDR|RejoinType|NetID|DevEUI|RJcount3|DeviceCompressedPublicKey)$$

$$MIC = cmac[0...3]$$

Notice that the `RJcount3` is a counter that is incremented at each *Rejoin-Request of type 3*, it shall never wrap around, due to the lifecycle of the device for a given `NwkKey` and `AppKey` value. The `RJcount3` prevents replay attacks for the given type of the Rejoin-Request since the Join/Network Server will discard all the *Rejoin-Request messages* with a counter value less or equal to the last valid *Rejoin-Request message of type 3* received. The `RJcount3` restarts from 0 after a successful type 3 Rejoin, so it has to be unique for a given value of `NwkKey` and `AppKey`.

**Table 5.** Rejoin-Request type 3 packet format.

| Rejoin Type = 3 | NetID | DevEUI | RJcount3 | End-Device Public Compressed Key |
| --- | --- | --- | --- | --- |
| 1 Byte | 3 Bytes | 8 Bytes | 2 Bytes | 33 Bytes |

### 5.2. Complete the Root Key Refresh Mechanism Using Join-Accept Message of Type 1

As soon as the Join/Network Server receives a valid *Rejoin-Request message of type 3*, it will generate a new pair of public/private ECC keys using the *secp256r1* curve. The Join/Network server will now compute a secret using the public key of the end-device and its private key using the ECDH protocol. It will transmit a down-stream *Join-Accept message of type 1* with the following fields: the `JoinNonce`, the `Home_NetID`, the `DevAddr`, the `DLSetting`, the `RxDelay`, and the *compressed public key* of the Join/Network Server.

The contents of the new packet type are depicted in Table 6. The *MIC* of the *Join-Accept message of type 1* is computed as follows:

$$cmac = aes128\_cmac(JSIntKey,$$

$$RejoinRequestType|JoinEUI|RJcount3|MDHR|JoinNonce|NetiD|DevAddr|$$

$$DLSetting|RxDelay|ServerCompressedPublicKey)$$

$$MIC = cmac[0...3]$$

The *Join-Accept message of type 1* will be encrypted as follows:

$$aes128\_decrypt(JSEncKey, JoinNonce|NetID|DevAddr|DLSettings|RxDelay|$$

$$ServerCompressedPublicKey|MIC|pad_{16})$$

where $pad_{16}$ is a pad of 3 bytes to make the *Join-Accept message* 48 bytes long, multiple of 16. It is used the AES decrypt operation in ECB mode to encrypt the message, the same as the actual *Join-Accept message*.

**Table 6.** Join-Accept type 1 message.

| JoinNonce | Home_NetId | DevAddr | DLSettings | RxDelay | Server Public Compressed Key |
|---|---|---|---|---|---|
| 3 Bytes | 3 Bytes | 4 Bytes | 1 Byte | 1 Byte | 33 Bytes |

The end-device, after receiving the *Join-Accept message of type 1*, can also compute the same secret using its private key and the public key of the Network Server. Since both the end-device and the Network Server have a common 256-bit secret, the 128 most significant bits will be used as new `NwkKey`, meanwhile the 128 less significant bits as new `AppKey`. The derivation of the new root keys is depicted in Figure 2.

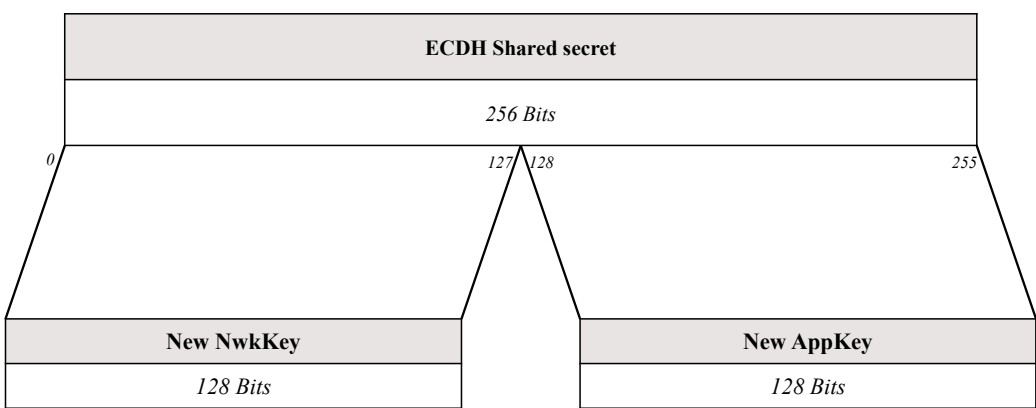

**Figure 2.** New Root Keys derivation.

After the generation of the new root keys both the end-device and the Network Server can discard the private/public keys previously generated, and they can recompute all keying material: `JSIntKey`, `JSEncKey`, `FNwkSIntKey`, `SNwkSIntKey`, `NewSEncKey` and `AppSKey`.

In case the end device faces a restart failure, the device needs to reissue a new Join-Request as specified in the actual specification of LoRaWAN1.1, but the two parties will use the newly computed root keys to generate the session and integrity keys.

## 6. Security Analysis

The most important requirement for LPWANs is the confidentiality and integrity of the data transmitting by IoT devices. Therefore, key distribution is critical for the protection of LPWAN networks and the prevention of adversaries from attacking the network

and potentially compromising the privacy and integrity of the data. The capabilities and the constraints of IoT devices' hardware influence the type of security mechanisms and protocols that can be hosted on the device's hardware. Moreover, the long-range wireless networking topology makes it susceptible to link attacks ranging from passive eavesdropping to active interference. For these reasons, the choice of a key establishment protocol for the creation of a shared, secret key must be done very carefully and should exhibit the following critical properties:

- Key authentication—assuring only intended IoT devices can access a key,
- Integrity—ensuring that there is no unauthorized data modification and
- Confidentiality—by providing security measures to avoid eavesdropping.

Given that IoT devices are designed to operate for very long periods, in many application scenaria such as Smart Metering, for periods longer than ten years, it is important to refresh the keys periodically while at the same time guaranteeing the above properties. It is, therefore, important to also consider *Key Freshness* as a minimum requirement for LPWANs [20–22]. It is, therefore, important to also consider the following cryptographic properties:

- Computational key secrecy—it must be computationally infeasible for any passive adversary to discover any key.
- Decisional key secrecy—there must be no information leaked other than public key information.
- Key independence—a passive adversary that knows a subset of keys must not discover any other information of the remaining keys. This property decomposes into:

  - Forward Secrecy—a passive adversary that knows a subset of keys must not discover any subsequent keys.
  - Backward secrecy—a passive adversary that knows a subset of keys must not discover any preceding keys.

LoRaWAN1.1 current specification respects these requirements and properties with respect to the integrity and session keys (`FNwkSIntKey`, `SNwkSIntKey`, `NwkSEncKey`, `AppSKey`), but it does not guarantee key freshness in the case of the root keys (`NwkKey`, `AppKey`) and consequently for the `JSIntKey` and `JSEncKey` keys.

The secure rejoin mechanism presented here guarantees key freshness for all these keys within the aforementioned cryptographic properties. Consider that an adversary that has acquired access to a public key, and thanks to the Elliptic Curve Cryptography, it is computationally infeasible to retrieve the private keys of both parties [53] or the shared secret computed via *ECDH*. So *computational key secrecy* and *decisional key secrecy* are guaranteed.

Regarding *key independence*, consider that the new root keys are completely independent of the previous and the subsequent ones since the computation of new root keys is based on the execution of the Elliptic-curve Diffie-Hellman protocol. Moreover, previous pairs of private/public ECC keys are discarded and a new pair is generated from scratch at each Rejoin-Request, both from the server and the end device.

In respect to the vulnerabilities listed in Section 4 and in particular, the *Man In The Middle attack*, remark that a malicious node can eavesdrop on the Rejoin-Request message since it is not encrypted. So the following information are plainly accessible: the `NetID`, the `DevEUI`, the `RJcount3` (i.e., the `DevNonce`), and the *end-device public key*. However, the attacker cannot obtain any information from the Join-Accept type 1 message, since it is encrypted. Nevertheless, it can derive the `JoinNonce` as it was described in Section 4. Based on the information collected, the attacker will still be unable to guess the value of the new `NwkKey` and `AppKey` keys or the new session keys. Hence, the attacker will not be able to compute the value of the ECDH shared secret knowing only the public key of the end device.

As for the *Replay Attack*, in the Rejoin-Request type 3 method, the `RJcount3` is a counter that is incremented each time the Rejoin-Request type 3 is invoked. The Network

Server will discard any Rejoin-Request of type 3 messages received with a `RJcount3` value smaller or equal to the value of the last Rejoin-Request type 3 valid message, for a given value of the `NwkKey` and `AppKey`. When the Rejoin of type 3 concludes, the `RJcount3` value will restart from zero. In this way, the Replay Attack can be countered. Therefore, the value of the `RJcount3` counter substitutes the `DevNonce` in the session keys generation but more importantly it is used to prevent Replay Attacks. Remark however that the proposed mechanism is still vulnerable to a Replay Attack combined with a selective RF jamming attack.

## 7. Experimental Evaluation

The proposed mechanism is evaluated regarding its suitability for devices with restricted resources in terms of time and energy efficiency. The evaluation is based on a series of experiments carried out using real-world devices provided by the FIT IoT-LAB testbed [28]. In total, six different microcontrollers are selected that are based on the ARM Cortex-M family, microcontroller cores designed for a wide range of embedded applications:

ARM Cortex-M0—one microcontroller, the `nRF51422` by *Nordic Semiconductor* is implementing the ARMv6-M architecture providing a 16 MHz 32-bit RISC core, with up to 256 KB of Flash memory, and up to 32 KB of RAM. It also includes an 128-bit AES/ECB/CCM/AAR co-processor.

ARM Cortex-M0+—two microcontrollers, the `STM32L072CZ` by *ST Microelectronics* and the `ATSAMR21G18A` by *Atmel* operate a 32 MHz 32-bit RISC core and aim to be much more compatible with 8 bit and 16 bit processors, with reduced energy consumption of up to 30% compared to ARM Cortex-M0 cores. The Cortex-M0+ has complete instruction set compatibility with the Cortex-M0 thus allowing the use of the same compiler and debug tools. The `STM32L072CZ` provides 192 KB of Flash program memory and 20 KB of RAM, while the `ATSAMR21G18A` provides 256 KB of Flash program memory and 32 KB of RAM. Both microcontrollers offer hardware support for 128-big AES operations.

ARM Cortex-M3—one microcontroller, the `STM32F103REY` by *ST Microelectronics* is the ARMv7-M architecture providing a 32-bit RISC core operating at a 72 MHz frequency, equipped with 64 KB of RAM and 256 KB of ROM. This microcontroller offers a performance of 1.25 DMIPS/MHz with a 3-stage pipeline, multiple 32-bit busses, clock speeds up to 200 MHz and very efficient debug options. The `STM32F103REY` does not provide any hardware support either for AES operations or for Pseudo-Random Number Generation.

ARM Cortex-M4—one microcontrollers, the `nRF52832` by *Nordic Semiconductor* is based on the ARM Cortex-M4 design that builds on top of Cortex-M3 by providing DSP instructions. It operates a 32-bit RISK core at 64 MHz and provides 512 KB of Flash memory and 64 KB of RAM and is equipped with a 128-bit AES/ECB/CCM/AAR co-processor.

ARM Cortex-M4F—one microcontrollers, the `nRF52840` by *Nordic Semiconductor* is based on the ARM Cortex-M4 design also a Floating-Point Unit. It operates a 32-bit RISK core at 64 MHz and provides 1 MB of Flash memory and 256 KB of RAM. Moreover, the 128-bit AES/ECB/CCM/AAR co-processor it is also equipped with an ARM TrustZone CryptoCell 310 security subsystem that provides a Pseudorandom number generator (PRNG) as well as support for Elliptic curve cryptography (ECC) and in particular, the SEC 2 recommended curve *secp256r1* using pseudorandom parameters, up to 521 bits.

These six microcontrollers are targeting a broad range of IoT applications, ranging from advanced wearable devices to interactive entertainment devices, home automation and industrial cyber-physical systems. For example, the *STM32F103REY* is used within the Apple TV 4 remote control. The selected set allows evaluating the performance over different combinations of resource constraints in terms of computational power (e.g.,

16 MHz of the `nRF51422`), limited memory (e.g., 20 KB of RAM by the `STM32L072CZ`), the availability of hardware support for 128-bits AES or the availability of hardware support for Pseudo-Random Number Generation and the *secp256r1* ECC curve. The specifications of the selected microcontrollers are summarized in Table 7.

**Table 7.** Overview of microcontroller specifications used in the experimental evaluation.

| Microcontroller | Architecture | Frequency | RAM | Hardware Support | | |
| :---: | :---: | :---: | :---: | :---: | :---: | :---: |
| | | | | AES | PRNG | ECC |
| nRF51422 | Cortex-M0 | 16 MHz | 20 KB | YES | NO | NO |
| STM32L072CZ | Cortex-M0+ | 32 MHz | 20 KB | YES | NO | NO |
| ATSAMR21G18A | Cortex-M0+ | 32 MHz | 32 KB | YES | NO | NO |
| STM32F103REY | Cortex-M3 | 72 MHz | 64 KB | NO | NO | NO |
| nRF52832 | Cortex-M4 | 64 MHz | 64 KB | YES | NO | NO |
| nRF52840 | Cortex-M4F | 64 MHz | 256 KB | YES | YES | YES |

In software, the proposed mechanism is implemented using the *RIOT Operating System* [26] since it supports many different architectures for 8 bit, 16 bit, 32 bit and 64 bit processors, provides a simple process manager with support for multi-threading, provides a generic network stack and also power management [54]. RIOT OS incorporates the `micro-ecc` library [27] that implement ECDH and ECDSA for 8-bit, 32-bit, and 64-bit processors. The implementation of the 128-bits AES encryption was based on the `crypto` module provided by *RIOT OS* [55]. These cryptographic function can be used within security protocols at the system level by providing seamless crypto support across software and hardware components [56].

A comprehensive resource analysis for widely used cryptographic primitives across different off-the-shelf IoT platforms, and quantify the performance impact of crypto-hardware is carried out in [57]. Interestingly, the results reported in [57] indicate that RIOT, a multi-purpose operating system for IoT deployments, integrating crypto-libraries, achieves a time and energy efficiency very close to having the code running directly on the microcontrollers, as reported in [58,59].

### 7.1. Time Efficiency

The first part of the experimental evaluation looks into the running times of the operations used by the proposed mechanism to generate the public and private keys using ECC and compressing/decompressing the keys. The running times of each operation for each different microcontroller are presented in Table 8. For comparison with AES, the table also includes the running time for the encryption of a single 128-bits AES block of plain text. The measurements are conducted using the RIOT OS timer module that uses the hardware RTC running at 32,768 Hz and provides a precision down to 30 µs. To obtain good average results, the operations were repeated at least 100 times.

Overall, the experimental results indicate the feasibility of the proposed secure rejoin mechanism in resource constraint devices. The computation of the EC-DH secret and the generation of the public and private keys are an order of magnitude slower than the other operations considered. These operations are implemented within the `micro-ecc` library and are based on multiplications of random numbers with a point on the elliptic curve that requires operations on 64 bytes long variables. The running time is therefore directly connected to the hardware design of the microcontroller in terms of memory access times. Nevertheless, even for the case of the `nRF51422` microcontroller that operates a 16 MHz ARM Cortex-M0 core, the required time averages out to 2.09 s. For the case of ARM Cortex-M3 and M4 microcontrollers considered in this study, this time is significantly reduced to about 350 ms.

For the case of the `nRF52840` microcontroller, one would expect that the availability of the external crypto-chip would improve the running times. The experimental results indicate that the running times are almost identical to those acquired from the `nRF52832`

microcontroller where no such external board is available. Once again this can be partially justified by the hardware design and the way the peripheral crypto module is interfaced with the microcontroller in terms of data access and control overhead. Remark that similar observations are reported in [57].

Taking into account that the proposed Rejoin mechanism needs to be executed at low frequencies, the overhead in terms of the running time is acceptable concerning the enhancement of the overall security of the LoRaWAN1.1 Over-The-Air Activation method.

**Table 8.** Time consumption of each ECC operations and the encryption of a single 128-bits AES block (in microseconds).

| Operation | nRF51422 | STM32L072CZ | ATSAMR21G18A | STM32F103REY | nRF52832 | nRF52840 |
|---|---|---|---|---|---|---|
| Compute ECC Pub/Priv Keys | 1,045,543 | 523,432 | 387,376 | 184,052 | 161,234 | 161,279 |
| Compress ECC Public Key | 43 | 20 | 22 | 12 | 10 | 10 |
| Decompress ECC Public Key | 79,216 | 39,639 | 29,298 | 13,881 | 11,778 | 11,754 |
| Compute EC-DH Secret | 1,045,496 | 523,353 | 387,330 | 184,035 | 161,253 | 161,245 |
| 128-bit AES encryption | 216 | 116 | 97 | 48 | 41 | 41 |

*7.2. Energy Efficiency*

Given the above running times for performing the necessary cryptography operations, the evaluation looks into the performance of the mechanism in terms of energy consumption. The evaluation of the energy consumption of each microcontroller is based on the Consumption Monitor of the FIT IoT-LAB testbed [28] that is available only for the ATSAMR21G18A and the STM32F103REY microcontrollers. This is because the consumption monitor is based on an external board that is attached to the microcontroller and measures the consumption through an INA226 hardware component of *Texas Instruments*. The device monitors both a shunt voltage drop and bus supply voltage using a programmable calibration value, conversion times and averaging. The conversion times (CT) for these measurements can be selected from as fast as 140 μs to as long as 8.244 ms. The conversion time settings, along with the programmable averaging mode (AV), allow the INA226 to be configured to optimize the available timing requirements in a given application. In the experiments reported here, the consumption monitor is configured to convert a filtered signal every 204 μs with the averaging mode set to 10 to have a periodic measure of 4.08 ms. Remark that a greater number of averages enables the INA226 to be more effective in reducing the noise component of the measurement.

The energy efficiency of the ECC operations used by the proposed secure rejoin mechanism are listed in Table 9. The table also includes the energy consumption when the microcontroller is in an idle state and also the energy consumption for the encryption of a single 128-bits AES block of plain text. The experiments indicate that the overall consumption of energy for the crypto mechanisms is comparable to the consumption during the idle state or for performing encryption using the AES module. The ATSAMR21G18A chip that is based on the ARM Cortex-M0+ design achieves higher energy efficiency than the STM32F103REY chip that follows the ARM Cortex-M3 design.

Like in the case of time efficiency, in the case of energy efficiency it seems that the overall overhead incurred by the rejoin mechanism is acceptable given the low frequency of execution and the improvements to the overall security of the LoRaWAN1.1 OTAA method.



**Table 9.** Power consumption of each ECC operations, the encryption of a single 128-bits AES block and during the idle state (in Watt).

| Operation | ATSAMR21G18A | STM32F103REY |
|---|---|---|
| Idle State | 0.056 W | 0.5200 W |
| Compute ECC Public/Private Keys | 0.130 W | 0.542 W |
| Compress ECC Public Key | 0.058 W | 0.5210 W |
| Decompress ECC Public Key | 0.129 W | 0.5237 W |
| Compute EC-DH Secret | 0.129 W | 0.5420 W |
| 128-bit AES encryption | 0.068 W | 0.5230 W |

## 8. Conclusions and Future Work

Emerging Low-Power Wide Area Network technologies can deliver scalable and long-term IoT deployments within urban environments or provide long-range coverage in rural environments. Such large-scale deployments require very careful consideration of all possible technical and security aspects as well as the establishment of users' trust. In this paper, the network initialization mechanism of LoRaWAN is examined, an LPWAN technology developed by the LoRa Alliance as an open standard that can operate over private and/or public infrastructures spreading all over the world.

Under the latest LoRaWAN specifications v1.1 (LoRaWAN1.1), the network initialization relies on two root keys, the `NwkKey` and the `AppKey`, which are hardcoded into the device firmware and remain the same for the entire duration of the network deployment. The security of the current specifications are evaluated against various active or passive attacks for the particular period where the device remains active. An attacker that manages to guess these keys can significantly impact the confidentiality of the messages and integrity of the network for the entire duration of the deployment. Taking into consideration that LPWANs are expected to operate for at least ten years, where devices need to be operational in remote locations, this is an important vulnerability.

In this paper, a secure method to refresh the root keys is proposed that can be executed at any time throughout the operation of an LPWAN. The method is using *Elliptic Curve Cryptography* to enable the secure exchange of the new root keys. The method proposed is taking advantage of the asymmetric encryption with an affordable overhead in terms of computational power, energy and time consuming, fitting particularly well in resource-constrained LoRaWAN devices. Moreover, the proposed method does not require any changes to the core architecture of LoRaWAN1.1 as it builds upon the existing Rejoin-Request message format. In this way backwards compatibility is guaranteed. The length of both the new type of messages introduced, even if they are bigger than the other messages used by the existing mechanism, they still fit within the maximum payload size defined by LoRaWAN1.1 that range from 51 to 222 bytes depending on the spreading factor, the frequencies and the bandwidth [60].

The security properties of the method are examined against a specific type of attacks. The analysis indicates that the new secure rejoin mechanism guarantees (i) computational key secrecy, (ii) decisional key secrecy, and (iii) key independence, both forward and backward secrecy, for both root keys. In respect to the Man in the Middle and Replay attacks considered, a malicious node that manages to eavesdrop on a Rejoin-Request message will still not be able to guess the value of the new root keys or the new session keys. Remark however that the proposed mechanism is still vulnerable to a Replay Attack combined with a selective RF jamming attack.

Apart from the analysis of the security properties of the proposed method, the method is implemented in software using the RIOT OS. The resulting software is evaluated regarding its suitability for devices with restricted resources in terms of time and energy efficiency. The evaluation is based on a series of experiments carried out using real-world devices provided by the FIT IoT-LAB testbed. In total, six different microcontrollers are selected that are based on the ARM Cortex-M family, over different combinations of re-

source constraints in terms of computational power, limited memory, the availability of hardware support for 128-bits AES or the availability of hardware support for Pseudo-Random Number Generation and the *secp256r1* ECC curve. These six microcontrollers are targeting a broad range of IoT applications, ranging from advanced wearable devices to interactive entertainment devices, home automation and industrial cyber-physical systems. The results indicate that the agreement of the new root keys requires a very short time and incurs a very limited energy consumption over a heterogeneous set of microcontrollers. The experiments indicate that the overall overhead incurred in terms of energy and time by the proposed rejoin mechanism is acceptable given the low frequency of execution and the improvements to the overall security of the LoRaWAN1.1 OTAA method.

As future work, it is important to continue the security analysis of LoRaWAN1.1 in terms of replay attacks that are combined with a selective RF jamming technique and also look into other types of attacks that may compromise network security.

**Author Contributions:** S.M. and I.C. have read and agreed to the published version of the manuscript. All authors have read and agreed to the published version of the manuscript.

**Funding:** This research was partially funded by the European Union's research project "Secure and Seamless Edge-to-Cloud Analytics" (ELEGANT), funded by the European Commission (EC) under the Horizon 2020 framework and contract number 957286. This document reflects only the authors' views and the EC is not responsible for any use that may be made of the information it contains.

**Data Availability Statement:** Not applicable.

**Conflicts of Interest:** The authors declare no conflict of interest.

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
