# Peer review of "Design, Analysis, and Experimental Evaluation of a New Secure Rejoin Mechanism for LoRaWAN Using Elliptic-Curve Cryptography"

_jsan, doi:10.3390/jsan10020036_

Round 1

Reviewer 1 Report

The authors report a security vulnerability to OTAA of LoRaWAN and provide a possible solution that is compatible with the existing version. Then, the operation of devices under standard version of OTAA is not limited whereas the security increases. The comparison with previous methods, as in lines 171 to 176 results interesting.

Along the paper there are some issues I think they must be solved.

I feel that the proposal is introduced after long introductory text (in fact, it is proposed at page 9 (of 16, as  the rest are references): I'm not completely sure if all these previous sections are really needed or most of the contents could be referenced to the standards or white papers.

Figures 1, 2, 4, 6, and 7 must be tables.

I suggest to improve the comparisons and the conclusions, as the novelty of the proposal is not clearly highlighted.

There are some sentences really difficult to understand. Please rewrite. Some examples:

  • lines 8 to 10
  • lines 93 to 94

Author Response

First of all we would like to thank the reviewer for the time and effort dedicated to go through our paper, examine our work and provide feedback on how to improve it. Regarding the specific comments, here is how we addressed them by revising the article:

Comment #1: "I feel that the proposal is introduced after long introductory text (in fact, it is proposed at page 9 (of 16, as the rest are references): I'm not completely sure if all these previous sections are really needed or
most of the contents could be referenced to the standards or white papers."

We have reduced the introduction and also the section presenting the LoRaWAN protocol. We believe that the information that is still in the paper is valuable for the reader that is not fully aware of the inner mechanisms of LoRaWAN1.1 and in this way makes the content more complete.

We would also like to note that section 4 "Security Issues & Vulnerabilities of LoRaWAN1.1 Over-The-Air-Activation" is also a new result. Therefore, taking also into account the modifications made, the new results start on page 7.

Comment #2: "Figures 1, 2, 4, 6, and 7 must be tables."

We converted the indicated figures to tables.

Comment #3: "I suggest to improve the comparisons and the conclusions, as the novelty of the proposal is not clearly highlighted."

We have significantly extended the final section to better highlight the novelty of the proposal.

Comment #4: "There are some sentences really difficult to understand. Please rewrite. Some examples: lines 8 to 10, lines 93 to 94"

We have rewritten these lines and also went through the entire article making corrections on the english language.

Reviewer 2 Report

Dear Authors,

The paper deals with a new method to improve the security of the LoRaWAN communications, by using a new way to generate session keys over the current hard-coded keys. Under the latest LoRaWAN specifications v1.1 (LoRaWAN1.1), OTAA uses two root keys, the NwkKey and the AppKey, to compute the session keys. While OTAA allows refreshing the session keys, it is not possible to refresh the aforementioned root keys. These two keys are hard-coded into the device firmware and remain the same for the entire duration of the network deployment.

The new mechanism proposed in this paper introduces a new type of Rejoin-Request and Join-Accept message that allow to refresh the root keys using Elliptic-Curve Cryptography in a way that guarantees computational key secrecy, key secrecy, key independence and key freshness.

All paper sections are very well documented and linked with the most current research results in the field. The experimental part presents encouraging results that could serve for real life implementations.

Overall it's a very good paper. Congratulations!

Author Response

We would like to thank the reviewer for the time and effort dedicated to go through our paper and examine our work.

We have went through the entire document and improved the english language by simplifying sentences and fixing various minor errors.

Reviewer 3 Report

The authors propose an increment of LoRaWAN OTAA security based on ECC root keys derivation (ReJoin).
Hence, the research topic is relevant.

I suggest reducing the abstract to 12-15 lines.

Many acronyms were not defined before its used.

It is essential to conduct an extensive grammatical revision. E.g., line 255 - 3.4 - "Refress" -> Refresh; line 266 - "is similar to to type 0"

Section 4 can be incorporated into another one and reduced. For example, merging sections 4 and 5.

It is essential to keep clear to the reader the difference between symmetric key cryptography schemes, like AES used by LoRaWAN 1.1, in contrast to asymmetric schemes, like RSA with ECC.

Proposing the use of ECDH on LoRaWAN ReJoin is relevant and interest, but it is not clear how LoRaWAN can implement ECDH without changing the current cryptographic support. Moreover, LoRaWAN devices are majority constrained-resource, then, it is necessary to define clearly the scope of LoRaWAN devices that can run the proposal. 

Author Response

We would like to thank the reviewer for the valuable comments and suggestions for improvement.

We created an revised version of our manuscript that attempts to address all the comments made by the reviewer. Here is a detailed explanation of the changes made:

Comment #1: "I suggest reducing the abstract to 12-15 lines."

We attempted to reduce the abstract, however due to some comments made by one of the other reviewers, we had to include some additional details. We hope that the new version of the abstract is sufficiently short.

Comment #2: "Many acronyms were not defined before its used."

We went through the paper and defined all the acronyms before their first usage. We hope that we did not miss any acronym.

Comment #3: "It is essential to conduct an extensive grammatical revision. E.g., line 255 - 3.4 - "Refress" -> Refresh; line 266 - "is similar to to type 0""

We went through the paper and conducted an extensive grammatical revision. We hope that we have not missed any error.

Comment #4: "Section 4 can be incorporated into another one and reduced. For example, merging sections 4 and 5."

In order to address one of the comments made by another reviewer, we are kindly asking to keep the same structure. We also believe that section 4 helps the reader better understand the vulnerabilities of the existing solution and in this way highlights the benefits of the proposed solution.

Comment #5: "It is essential to keep clear to the reader the difference between symmetric key cryptography schemes, like AES used by LoRaWAN 1.1, in contrast to asymmetric schemes, like RSA with ECC."

We have included an explanation in the introductory section of the paper at the point where the results are introduced to the reader.

Comment #6: "Proposing the use of ECDH on LoRaWAN ReJoin is relevant and interest, but it is not clear how LoRaWAN can implement ECDH without changing the current cryptographic support. "

We have included an additional explanation in the introductory section explaining that no changes to the architecture or the message structures of LoRaWAN1.1  are needed to support the proposed solution. We also mention that devices that do not implement the proposed solution will continue to operate without any disruption.

Comment #7: "Moreover, LoRaWAN devices are majority constrained-resource, then, it is necessary to define clearly the scope of LoRaWAN devices that can run the proposal. "

We have included an additional explanation in the introductory section and the abstract explaining that the microcontrollers used in the experimental evaluation are using the ARM Cortex-M architecture also including ultra-low power M0+ cores. In this way we hope that it is clear which devices can support the the proposed scheme. We believe that this is a very broadly used microcontroller architecture in modern IoT deployments.

Round 2

Reviewer 1 Report

My previous comments have been addressed, so I have no additional concerns.

Author Response

We would like to thank the reviewer for the valuable comments and suggestions for improvement.